# Canada Goldenrod Invasion Regulates the Effects of Soil Moisture on Soil Respiration

**DOI:** 10.3390/ijerph192315446

**Published:** 2022-11-22

**Authors:** Sixuan Xu, Kexin Li, Guanlin Li, Zhiyuan Hu, Jiaqi Zhang, Babar Iqbal, Daolin Du

**Affiliations:** 1School of Environment and Safety Engineering, Jiangsu University, Zhenjiang 212013, China; 2Key Laboratory of Environmental Biotechnology, Research Center for Eco-Environmental Sciences, Chinese Academy of Sciences, Beijing 100085, China; 3Institute of Environmental Health and Ecological Security, School of Environment and Safety Engineering, Jiangsu University, Zhenjiang 212013, China; 4WM Environmental Molecular Diagnosis Co., Ltd., Suzhou 215558, China; 5Ministry of Education Key Laboratory for Ecology of Tropical Islands, Key Laboratory of Tropical Animal and Plant Ecology of Hainan Province, College of Life Sciences, Hainan Normal University, Haikou 571158, China

**Keywords:** alien plant invasion, invasive effect, soil autotrophic respiration, soil heterotrophic respiration

## Abstract

Canada goldenrod (*Solidago canadensis* L.) is considered one of the most deleterious and invasive species worldwide, and invasion of riparian wetlands by *S. canadensis* can reduce vegetation diversity and alter soil nutrient cycling. However, little is known about how *S. canadensis* invasion affects soil carbon cycle processes, such as soil respiration, in a riparian wetland. This study was conducted to investigate the effects of different degrees of *S. canadensis* invasion on soil respiration under different moisture conditions. Soil respiration rate (heterotrophic and autotrophic respiration) was measured using a closed-chamber method. *S. canadensis* invasion considerably reduced soil respiration under all moisture conditions. The inhibition effect on autotrophic respiration was higher than that on heterotrophic respiration. The water level gradient affects the soil autotrophic respiration, thereby affecting the soil respiration rate. The changes in soil respiration may be related to the alteration in the effective substrate of the soil substrate induced by the invasion of *S. canadensis*. While the effects of *S. canadensis* invasion were regulated by the fluctuation in moisture conditions. Our results implied that *S. canadensis* invasion could reduce the soil respiration, which further potentially affect the carbon sequestration in the riparian wetlands. Thus, the present study provided a reference for predicting the dynamics of carbon cycling during *S. canadensis* invasion and constituted a scientific basis for the sustainable development and management of riparian wetlands invaded by alien plants.

## 1. Introduction

With economic globalization, alien plant invasion has become an important ecological problem. Numerous studies found that invasive plants typically show high net primary productivity and can change aboveground vegetation community structures, thus affecting the quantity and quality of soil carbon input [1,2], and subsequent changes in soil respiration may affect carbon pools (e.g., regarding carbon sequestration and carbon input) [3,4,5]. Therefore, plant invasion markedly affects the global carbon cycle.

Wetland ecosystems constitute an important carbon pool and play an important role in carbon storage. However, wetlands are fragile ecosystems that are vulnerable to invasion by alien plants. Invasion of the Hangzhou Bay wetlands by Canada goldenrod (*Solidago canadensis* L.) considerably reduced the soil pH and changed the organic carbon components [6], and invasion of coastal wetlands of eastern China by *Spartina alterniflora* has changed the biodiversity and carbon poverty of coastal wetland ecosystems [7]. Thus, invasion by alien plants affects community diversity, microbial activity, and soil physical and chemical properties of wetland ecosystems [8].

Soil respiration is an important component of the carbon cycle in terrestrial ecosystems, and it is the main CO_2_ output from the soil carbon pool to that of the atmosphere. This complex biochemical process includes autotrophic (root respiration and root microbial respiration) and heterotrophic respiration (microbial and animal respiration) [9], and can be affected by the structure and activity of soil biomes, organic matter, and vegetation, as well as the soil physicochemical characteristics, such as soil temperature, moisture, nutrients, and pH [10,11]. Among them, soil temperature and moisture are the main factors affecting soil respiration, which in turn is affected by vegetation structure [12]. However, the potential impact of alien plant invasion on soil respiration is so far unclear. Previous studies found inconsistent effects of invasive plants on soil respiration. Invasion of wetlands by *Acacia farnesiana* (Linn.) Willd. and *Cynara cardunculus* (L.) increased soil respiration [13,14], and invasion of coastal wetlands by *Spartina alterniflora* was predicted to markedly increase greenhouse gas emissions [15]. However, this invasion by *Spartina alterniflora* was shown to reduce soil respiration, which was contrary to other observations, likely because soil respiration depends on the content and input quality of soil total organic carbon, the decomposition process of litter, or the fluctuation of the groundwater level in wetlands [16,17,18,19]. This suggests that the alien plant invasion effects on the soil carbon cycle depend on various biotic and abiotic factors [14,20,21,22,23,24,25]. The effects of soil moisture on respiration are complex. Under low soil moisture, soil respiration is strongly correlated with soil moisture [26], whereas soil respiration peaks near field capacity. Beyond a certain threshold, soil respiration may decline, and when soil moisture is saturated, soil respiration stops [27,28].

*Solidago canadensis* L.is native to North America and is currently considered one of the most deleterious and pervasive invasive species worldwide [29]. In 1913, *S. canadensis* was introduced to Shanghai, China as ornamental flowers. Subsequently, *S. canadensis* spread to natural environment, including the area south of the Yangtze River, and has become one of the most harmful weeds in China [30,31]. The impact of *S. canadensis* on soil respiration have rarely been examined [32]. Previous studies have found that the invasion of the riparian wetlands by *S. canadensis* affected soil respiration and the carbon cycle in the invasion area, and with the intensification of the invasion, soil respiration showed a decreasing trend [33,34]. This inhibition of soil respiration might be caused by the changes in the underground soil microenvironment and aboveground vegetation community structure under *S. canadensis* invasion. Invasion by *S. canadensis* affects autotrophic and heterotrophic respiration, but the contribution of these two components to soil total respiration is unclear. Moreover, soil moisture is important to consider in this regard, however, the specific interaction effect of alien plant invasion and soil moisture conditions on soil respiration remains to be elucidated [35].

Thus, the present study was conducted to investigate the effects of different degrees of *S. canadensis* invasion on soil respiration under different moisture conditions. We hypothesized that (1) *S. canadensis* invasion would inhibit soil respiration, as well as all components of soil respiration (autotrophic and heterotrophic respiration); and (2) the effect of invasion on these three types of respiration would depend on soil moisture.

## 2. Materials and Methods

### 2.1. Experimental Design

The experiment was conducted in a nursery at Jiangsu University (32°12′ N, 119°30′ E), Zhenjiang, China. *S. canadensis* invasion in a riparian wetland habitat (32°14′ N, 119°29′ E) of Zhenjiang was simulated. The originally predominant plant in this riparian wetland habitat was common reed (*Phragmites australis* (Cav.) Trin. ex Steud); however, this habitat has been invaded by *S. canadensis* in recent years.

Seeds of *P. australis* and *S. canadensis* were collected from a riparian wetland in December 2018. To preclude the *S. canadensis* invasion effects on soil characteristics, soil was collected from a non-*S. canadensis* invaded green space at the campus of Jiangsu University. The collected soils were sieved to remove stones and visible plant debris, and then placed in plastic pots (height: 26.5 cm, top diameter: 24.5 cm, and bottom diameter: 19.5 cm). After 2 months cultivation, similar size seedlings of *P. australis* and *S. canadensis* were carefully transplanted to pots in June 2019.

Invasion by *S. canadensis* was simulated by substituting space for time, i.e., different ratios of *P. australis* to *S. canadensis* were used to represent five successive stages of *S. canadensis* invasion, including the non-invasive (NI), early invasive (EI), intermediate invasive (II), dominant invasive (DI), and completely invasive (CI) stages, which have been previously described in detail [36]. Four seedlings were planted per pot, and different soil moisture conditions were simulated using water tanks. The pots were placed in tanks with different water levels, i.e., high, intermediate, and low, which were three-quarters, half, and one-quarter of the pot height, respectively. During the experiment period, the soil moisture among three water level treatments were significant difference (*p* < 0.01; Appendix A). All water tanks containing pots were placed in a nursery under natural light, and water was replenished every two days. The experiment was executed using a complete factorial design with three replicates (45 pots in total).

### 2.2. Soil, Autotrophic, and Heterotrophic Respiration Measurements

Soil respirations, including heterotrophic and autotrophic respiration, were measured between 08:00 AM and 10:00 AM on the first and fifteenth day of the month, from 15 July to 15 December 2019, by using a closed-chamber system. The introduction of the closed-chamber system and the method of soil respiration determination were described in detail in the previous study [37,38]. In brief, the chamber was directly inserted into the soil after removing the weeds, and the carbon dioxide content in the chamber was recorded every 5 s for 300 s. The topsoil (0–10 cm) temperature and moisture were measured using a soil temperature and moisture measurer (TR-6D, Shunkeda Technology Co., Ltd., Beijing, China). The root exclusion method, based on inserting deep gauze collars (diameter: 5.0 cm and height: 30 cm) at the same place of pots, was drawn upon to distinguish soil total respiration into heterotrophic and autotrophic respiration. Soil total, heterotrophic, and autotrophic respiration were calculated following the equations described in the previous study [34,37,38].

### 2.3. Soil and Vegetation Sample Collection and Preparation

Soil samples were collected by mixing topsoil (0–10 cm) obtained from different points along an X-shaped pattern in non-invasive, intermediate invasive, and completely invasive stage treatment pots among all water level treatments on 15 June 2019 (experiment start date) and 15 December 2019 (experiment end date). Each soil sample was divided into two parts, one of which was stored at 4 °C until soil microbial biomass and extracellular enzymatic activity analyses were performed, and the other was air-dried for soil characteristics analyses.

Plants were harvested on 15 December 2019. The harvested plants were weighed after 72 h oven-drying at 65 °C. Total biomass and root biomass in the non-invasive, intermediate invasive, and completely invasive stage treatment pots were calculated as the dry mass of all plant per pot.

### 2.4. Soil Characteristics

Soil pH was quantified in soil suspensions at a ratio of 1:5 (air-dried soil weight to deionized water volume). Soil dissolved organic carbon and nitrogen content were quantified using a total organic carbon analyzer with nitrogen module (Shimadzu TOC-L, Kyoto, Japan) [39]. Soil nitric nitrogen content was quantified following the colorimetric method [40,41]. Soil total carbon and soil total nitrogen content was quantified using an elemental analyzer (Vario MACRO; Elementar Analysensysteme GmbH, Langensebold, Germany). Soil total phosphorus content was measured using the molybdate colorimetry method [42,43]. Soil microbial biomass carbon, nitrogen, and phosphorus content was quantified using the chloroform fumigation extraction method [44,45]. Soil microbial community diversity (H’) was quantified using a BIOLOG EcoPlate (Biolog Inc., Hayward, Berkeley Heights, NJ, USA) following the measurement procedure described previously [46]. Extracellular activity of carbon-acquiring enzymes (β-D-1,4-cellobiohydrolase, β-1,4-xylosidase, and β-1,4-glucosidase), that of nitrogen-acquiring enzymes (L-leucine aminopeptidase and 1,4-N-acetylglucosaminidase), and that of the phosphorus-acquiring enzyme phosphatase were quantified following the procedure described by DeForest (2009) [47]. Microbial energy limitation (VL), microbial nutrient limitation (VA), and microbial carbon utilization (CUE) were calculated according to Cui et al. (2020) [48].

### 2.5. Statistical Analyses

Two-way analysis of variance (ANOVA) was used to test the individual and interaction effects of *S. canadensis* invasion and water level on soil total, autotrophic, and heterotrophic respiration and their respective effects on soil characteristics. The variations of soil characteristics between start date sampling and end date sampling were used. Analysis of covariance and linear regression tests were performed to determine the univariate relationship between soil respiration and soil moisture for each degree of *S. canadensis* invasion. Partial least squares path modeling (PLS-PM) was used to test the possible pathways by which the factors impact heterotrophic and autotrophic respiration of soil. All tests were executed using SAS version 9.4 (SAS Institute, Cary, NC, USA), except for the PLS-PM which was executed using Amos in IBM SPSS (version 24.0; SPSS Inc., Chicago, IL, USA).

## 3. Results

### 3.1. Effects of Treatments on Soil Total, Heterotrophic, and Autotrophic Respiration

*Solidago canadensis* L. invasion significantly affected soil total, heterotrophic, and autotrophic respiration (*p* < 0.01, each), and water level significantly affected soil total (*p* < 0.05) and autotrophic respiration (*p* < 0.05). However, no interaction effects were observed (Table 1). In general, the responses of soil total, heterotrophic, and autotrophic respiration to *S. canadensis* invasion showed similar trends under different water levels. With increasing *S. canadensis* invasion (from NI to CI), soil autotrophic respiration first decreased and then increased, whereas total soil respiration and autotrophic respiration showed a decreasing trend and were the lowest at the II stage. In particular, total soil respiration was lower than the initial value, whereas autotrophic respiration was disparate. Compared with the NI (only *P. australis* existence) treatment, at low water levels, *S. canadensis* growth reduced soil total respiration and heterotrophic respiration in the EI, II, DI, and CI stages by 14.3% and 18.0%, 31.7% and 32.7%, 13.1% and 33.6%, 8.5%, and 34.9%, respectively; soil autotrophic respiration was reduced by 4.7% and 32.0% in the EI and II treatments, but was increased by 36.2% and 60.6% in the DI and CI treatments, respectively (*p* < 0.01, each). At intermediate water levels, *S. canadensis* growth reduced soil total respiration and heterotrophic respiration in the EI, II, DI, and CI treatments by 19.2% and 12.7%, 34.2% and 30.6%, 24.9% and 29.7%, 14.8%, and 35.8%, respectively; soil autotrophic respiration in the EI, II, and DI treatments was reduced by 31.7%, 41.2%, and 15.7%, respectively, but was increased by 25.8% in the CI treatment (*p* < 0.01, each). At high water levels, *S. canadensis* growth reduced the soil total respiration and heterotrophic respiration in the EI, II, DI, and CI treatments by 9.7% and 18.8%, 28.8% and 22.0%, 6.3% and 22.0%, 7.5%, and 35.1%, respectively; soil autotrophic respiration was reduced by 19.5% and 42.4% in the EI and II treatments, but was increased by 18.2% and 36.8% in the DI and CI treatments, respectively (*p* < 0.01, each; Figure 1).

### 3.2. Correlations of Soil Total, Heterotrophic, and Autotrophic Respiration with Soil Characteristics and Plant Biomass

Soil moisture was negatively correlated with soil total, heterotrophic, and autotrophic respiration (*p* < 0.01, each). Covariance analysis showed an interaction effect of *S. canadensis* invasion and soil moisture on soil total respiration (Table 2). *S. canadensis* invasion altered the fitting curve between total respiration and moisture. With increasing invasion of *S. canadensis*, the adverse effect of increased soil moisture on soil total respiration decreased (Figure 2).

PLS-PM showed that *S. canadensis* invasion and water level affected soil nutrient availability, microbial characteristics, and plant biomass and ultimately affected soil total, heterotrophic, and autotrophic respiration. The total effects of plant root biomass (−0.418), CUE (−0.269), and alkaline phosphatase activity (−0.497) on soil total, heterotrophic, and autotrophic respiration were the highest. In addition, the direct driving factors differed between soil total, heterotrophic, and autotrophic respiration: root biomass and CUE were the direct driving factors of total soil respiration; CUE was a direct driving factor of soil heterotrophic respiration, whereas alkaline phosphatase activity and root biomass were those of soil autotrophic respiration (Figure 3).

## 4. Discussion

### 4.1. Effects of S. Canadensis Invasion on Soil Total, Heterotrophic, and Autotrophic Respiration

Compared with native plants *(P. australis)*, alien invasion plants generally demonstrate high growth rates, high biomass production, and strong reproductive capacity, which affect the quality and quantity of soil carbon input. Therefore, we initially hypothesized that *S. canadensis* invasion would promote soil respiration. However, the results showed that invasion by *S. canadensis* reduced soil respiration, but its effects on soil total, heterotrophic, and autotrophic respiration were different. At the early stage of invasion by *S. canadensis*, soil total, heterotrophic, and autotrophic respiration were inhibited, whereas at the DI stage, autotrophic respiration was promoted. In particular, autotrophic respiration was higher at the CI than at the EI stage (Figure 1). These results confirm the original hypotheses.

*Solidago canadensis* L. was found to have different driving mechanisms for soil total, autotrophic, and heterotrophic respiration [36,49]. In detail, *S. canadensis* can affect soil total respiration by affecting the root system of the invaded vegetation community and the composition and structure of the microbial community through changing the availability of soil substrates. *S. canadensis* invasion alters autotrophic respiration by interfering with root nutrient absorption and utilization and limiting root physiological activities [34]. Moreover, *S. canadensis* alters heterotrophic respiration by affecting the microbial decomposition of soil organic matter and litter through changing the composition, structure, and metabolic activity of the soil microbial community [33,50]. Allelochemicals including α-pinene, limonene, and germacrene, which are released by *S. canadensis* affect the availability of soil substrates [51,52]. However, these complex compounds also affect the soil microbial community structure, microbial metabolic limitations, and microbial nutrient utilization, which in turn alters microbial respiration [2]. This was confirmed by the observed changes in H’ (*p* < 0.01) CUE (*p* < 0.05) in the present study (Appendix A). PLS-PM revealed that *S. canadensis* invasion affected the soil microbial community and carbon availability by reducing carbon-related substrate availability, thereby inhibiting the activities of extracellular enzymes and microbial metabolism. The inhibiting effect may further force microbes to improve the utilization of carbon and reduce the release of CO_2_, which consequently suppresses microbial respiration (Figure 3). Therefore, the *S. canadensis* invasion effects on soil total, heterotrophic, and autotrophic respiration may be caused by changes in soil substrate availability.

### 4.2. Effects of Water Level on Soil Total, Heterotrophic, and Autotrophic Respiration

Water levels produced disparate effects on soil respiration under different degrees of *S. canadensis* invasion. Soil respiration decreased with increasing water levels (Figure 1). The water level effect on soil respiration was attributed to differences in vegetation biomass and growth rate, soil microbial activity, and soil nutrient availability. The effects of soil moisture on soil respiration are complex, and it is generally believed that there is a threshold for the effect of soil moisture on soil respiration, which typically depends on field capacity. Generally, the soil moisture effect on soil respiration can be divided into three situations: (1) below field capacity, soil respiration is positively correlated with soil moisture; (2) within a certain range, there is no pronounced relationship between soil respiration and moisture; and (3) above field capacity, soil respiration is negatively correlated with soil moisture. In the present study, a riparian wetland environment was simulated with soil moisture exceeding typical field capacity. Therefore, soil respiration showed a negative response to increased soil moisture.

The soil moisture effect on soil respiration was mainly reflected in three aspects. First, soil moisture is necessary for soil microbial activity and plant root photosynthesis. Under low or high soil moisture, protective mechanisms of plants and soil microbes may help mitigate adverse effects. For example, when soil moisture is too low, soil microbes transfer energy to produce appropriate nutrient solutes (carbon fixation), thus preventing adverse effects on plants and soil microbes. During this process, the release of CO_2_ decreases, thereby affecting soil respiration [27,28]. Second, soil moisture directly regulates the permeability of soil pores and oxygen [53]. The content and diffusivity of oxygen are reduced as soil moisture increases, thereby directly affecting the respiration of soil microbes and plant roots [6,54]. Excessive soil moisture inhibits the diffusion of oxygen in the soil, consequently limiting the growth of plant roots and reducing autotrophic soil respiration (root respiration) [55]. Meanwhile, the activity of aerobic microbes is also reduced, consequently affecting the decomposition of soil organic matter and the nutrient utilization pattern of microbes. These changes in soil microbes inhibit heterotrophic respiration (microbial respiration). The higher soil moisture in the current study affected soil autotrophic respiration and soil heterotrophic respiration by affecting plant roots and soil microbes. Third, soil moisture can change soil DOC, which is the main source of soil microbial activity energy. The enhancement of soil moisture can facilitate the diffusion of soluble organic carbon in the soil, which is convenient for microbes to absorb and utilize, thereby promoting microbial respiration [28,56]. However, these previous results are contrary to those of the present study. One possible explanation is that the promotion effect of higher moisture on the diffusion of DOC in the soil is reduced when a threshold of field water volume is exceeded (Appendix A) [57]. Soil moisture was previously suggested to affect the soil respiration process by altering the soil pH and soluble substance concentrations [58]. In the present study, changes in soil moisture reduced the soil pH and further affected plant growth (total biomass and root biomass) and the community composition and carbon use efficiency of microbes. In addition, changes in soil moisture also reduced the concentration of dissolved organic matter and affected the stoichiometric balance of soil nutrients (Appendix A). These alterations in plant and soil microbes affect soil respiration.

### 4.3. S. canadensis Invasion Affects Soil Respiration Responses to Soil Moisture

In the present study, soil moisture was significantly negatively correlated with soil total, heterotrophic, and autotrophic respiration, and *S. canadensis* invasion altered the negative correlation between soil moisture and soil respiration (Table 2; Figure 2). The invasion process of *S. canadensis* reduces the negative effect of the increase in soil moisture content on soil respiration, and complete invasion may gradually restore the adverse effects of increased soil moisture on soil respiration. It is possible that interspecific competition between native and invasive species intensifies the restriction of substrate availability on soil respiration when water is relatively abundant and reduces the negative effect of water-related factors to a certain extent [32,33]. This indicates that the impact of invasion by *S. canadensis* on soil respiration not only occurs through changes in substrate availability but also has a subsequent impact by affecting soil moisture.

With *S. canadensis* invasion, soil autotrophic respiration first decreased and then increased under each water level, and soil total respiration and heterotrophic respiration showed a continuous decreasing trend (Figure 1). In terrestrial ecosystems, soil respiration increases with increasing soil moisture [27], peaking near field capacity. This was inconsistent with the results of the present study, which may be because this study simulated a nearshore wetland system. Soil moisture was thus high (weight moisture content >30%) and exceeded field capacity, resulting in reduced soil respiration.

Soil autotrophic respiration represents the carbon flux produced by plant roots, mycorrhiza, and rhizosphere microorganisms [59]. As the invasion of *S. canadensis* increases net primary productivity, more carbon is distributed to aboveground plant organs, soil nutrients are limited, and autotrophic respiration is reduced. A previous study showed that autotrophic respiration is stimulated due to increasing soil water by increasing the carbon substrate supply and improving soil nutrient availability [60]. Under *S. canadensis* invasion, the aerenchyma of the root system expands. Under the influence of soil moisture, it is necessary to balance plant growth, nutrient and water absorption, and hormones, and high soil moisture conditions promote respiration of the plant root system [32]. This may explain why soil autotrophic respiration showed a decreasing trend first and then an increasing trend.

## 5. Conclusions

*Solidago canadensis* L. invasion significantly reduced soil respiration, and the inhibitory effect on autotrophic respiration was stronger than that on heterotrophic respiration. Water levels affected soil total respiration and autotrophic respiration. The changes in soil respiration may be related to the alteration in the effective substrate of the soil substrate induced by the invasion of *S. canadensis* and the fluctuation in moisture conditions. The change in soil substrate availability may not only affect the uptake and utilization of nutrients by plants and root physiological activities, but also affect soil heterotrophic respiration. As soil moisture can be used as a solvent and mobile carrier of soil nutrients, it could regulate the response of soil respiration to the invasion of *S. canadensis*. This study provides a reference for predicting the dynamics of the carbon cycle during the invasion process of *S. canadensis* and a scientific basis for the sustainable development and management of riparian wetlands invaded by alien plants.

## Figures and Tables

**Figure 1 ijerph-19-15446-f001:**
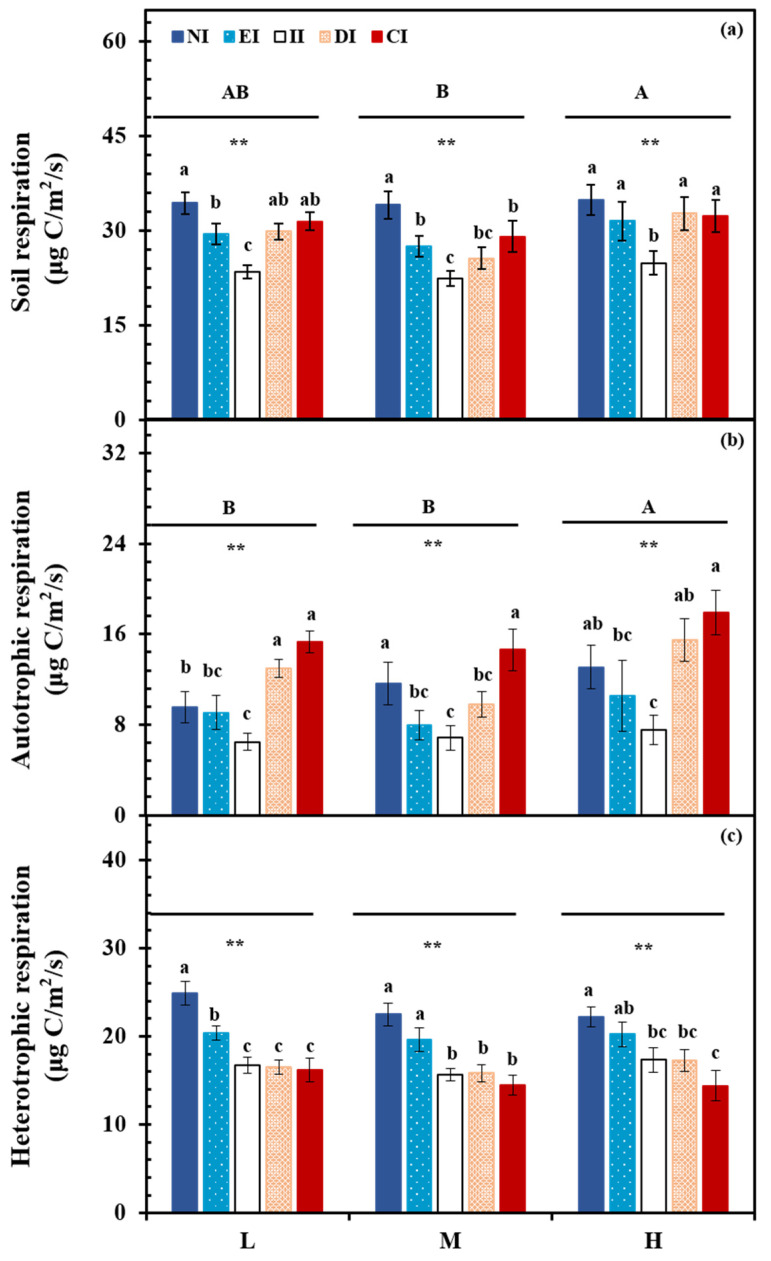
Soil (**a**), autotrophic (**b**), and heterotrophic (**c**) respiration under treatments. NI = non-invasive stage treatment, EI = early invasive stage treatment, II = intermediate invasive treatment, DI = dominant invasive treatment, CI = completely invasive stage treatment, L = low water level treatment, M = intermediate water level treatment, and H = high water level treatment. ** present significant at *p* < 0.01 among water level treatment, respectively. Different lower-case letters above column represent significant differences (*p* < 0.05) among invasive degree treatments.

**Figure 2 ijerph-19-15446-f002:**
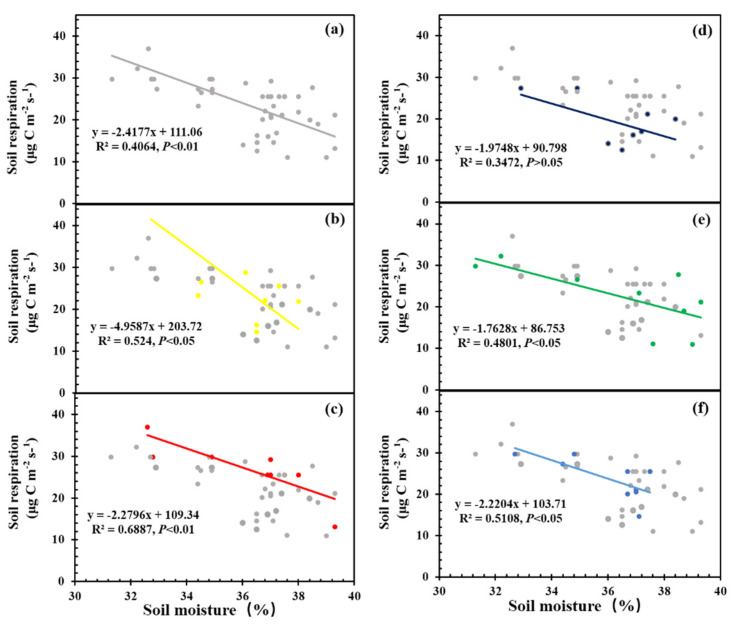
(**a**) Linear model of soil respiration and soil moisture under all invasive stages (gray points and line). (**b**–**f**) Linear model of soil respiration and soil moisture under non-invasive stage (yellow points and line), early invasive stage (red points and line), intermediate invasive stage (navy blue points and line), dominant invasive stage (green points and line), and completely invasive stage (light blue points and line).

**Figure 3 ijerph-19-15446-f003:**
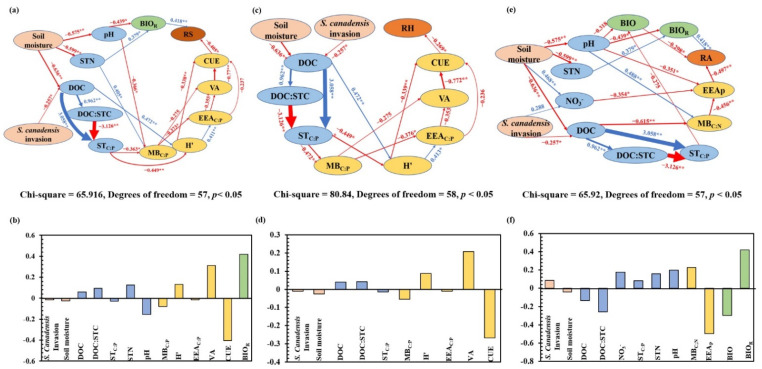
Partial least squares path modeling analysis of the effects of invasion and soil moisture on soil (RS; (**a**,**b**)), heterotrophic (RH; (**c**,**d**)), and autotrophic (RA; (**e**,**f**)), and the results of standardized total effects of various factors. DOC = dissolved organic carbon, STC = soil total carbon, DOC:STC = the ratio of DOC to STC, STN = soil total nitrogen, ST_C:P_ = the ratio of STC to soil total phosphorus, ${{\mathrm{NO}}_{3}}^{-}$ = nitrate nitrogen, MB_C:P_ = the ratio of microbial biomass carbon to microbial biomass phosphorus, H’ = soil microbial community diversity, EEA_C:N_ = the ratio of extracellular carbon-acquiring enzymic activity to nitrogen-acquiring enzymic activity, EEA_C:P_ = the ratio of extracellular carbon-acquiring enzymic activity to phosphorus-acquiring enzymic activity, VA = microbial nutrient limitation, CUE = microbial carbon utilization, BIO = plant biomass, and BIO_R_ = plant root biomass. * and ** present significant at *p* < 0.05 and *p* < 0.01, respectively.

**Table 1 ijerph-19-15446-t001:** Two-way ANOVA analysis of the individual and interactive effects of *S. canadensis* invasion and water level on soil total (RS), autotrophic (RA), and heterotrophic (RH) respiration.

	*S. canadensis* Invasion			Water Level			Interactive Effect		
	df	F	*p*	df	F	*p*	df	F	*p*
RS	5	38.16	** ^1^	2	3.52	*	10	0.52	0.87
RA	5	40.85	**	2	3.97	*	10	0.66	0.76
RH	5	23.33	**	2	1.48	0.23	10	0.44	0.92

^1^ * and ** present significant at *p* < 0.05 and *p* < 0.01, respectively.

**Table 2 ijerph-19-15446-t002:** Covariance analysis of the univariate relationship between soil (RS), autotrophic (RA), and heterotrophic (RH) respiration and soil moisture under *S. canadensis* invasion.

	*S. canadensis* Invasion			Water Level			*S. canadensis* Invasion * Soil Moisture		
	df	F	*p*	df	F	*p*	df	F	*p*
RS	3	0.26	0.62	3	11.92	** ^1^	9	7.8	**
RA	3	1.88	0.18	3	30.22	**	9	2.55	0.12
RH	3	2.86	0.09	3	19.16	**	9	3.11	0.08

^1^ * and ** present significant at *p* < 0.05 and *p* < 0.01, respectively.

## Data Availability

The datasets used or analyzed during the current study are available from the corresponding author on reasonable request.

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
