# Peer review of "Canada Goldenrod Invasion Regulates the Effects of Soil Moisture on Soil Respiration"

_ijerph, 2022, doi:10.3390/ijerph192315446_

Round 1

Reviewer 1 Report

The mansucript by Xu eat al studied the effects of plant invasion and water level on sol respiration, which emiited large amount of greenhouse gas  to the atmospheric environment. Considering the climate changing context and increase in atmoepheric greenhouse gas, this study provided up-to-date knowledge on management of invasive plants and evaluation of plant invasion effects on global change.

I have several suggestions and comments for further improvement of the manuscript, which could be considered by the authors, after which, it could be considered to publish in IJERPH.

Please see all comments and suggestions in the attached fils with notes and highlighted comments.

Author Response

Thank you for reviewing our manuscript and providing useful suggestions and comments to improve the quality of the manuscript. We considered all your suggestions and revised our manuscript.

Point 1: Abstract, lines -22- Soil respiration should be changed to soil respiration rate.

Response 1: Thank you for the comment. We have changed the “soil respiration” to the “soil respiration rate” in line 22. Now, this revised sentence reads as follows: “Soil respiration rate (heterotrophic and autotrophic respiration) was measured using a closed-chamber method. S. canadensis invasion considerably reduced all components of soil respiration under all moisture conditions.” (Line 22)

Point 2: Abstract, lines -25- If autotrophic respiration was affected then total Rs will be affected, too. This sentence could be more concise.

Response 2: Thank you for your comments. We rewrote the sentence succinctly and clearly according to your comments, which now reads as follows: “The water level gradient affects the soil autotrophic respiration, thereby affecting the total soil respiration.” (Line 25)

Point 3: Abstract, lines -26- As the only sentence related to mechanisms of goldenrod invasion effects on Rs, this sentence should present more data or evidence associated with the study.

Response 3: Thank you for your valuable comments. We have supplemented the relevant information to better understand the causes of changes in soil respiration. The rewritten sentences read as follows: “The changes in soil respiration may be related to the alteration in the effective substrate of the soil substrate induced by the invasion of S. canadensis. While the effects of S. canadensis invasion were regulated by the fluctuation in moisture conditions.” (Line 26)

Point 4: Introduction, lines -49- "and it is the main CO2 output from the soil carbon pool to the atmospheric carbon pool.” This sentence should delete the carbon pool and add "that of" before it.

Response 4: Thank you for the comment. We have revised this sentence, and now the revised sentences read as follows: “and it is the main CO2 output from the soil carbon pool to that of the atmosphere.” (Line 55)

Point 5: Introduction, lines -61- Including also factors like quality of SOC input, litter decomposition process (Zhang,et al., Scientific Reports 4: 05488; Zhang, et al., PLOS ONE9: e92301), or increase in groundwater levels in wet lands(Cui et al., Environmental Pollution, 286.2021,117400)?

Response 5: Thank you for your comment. According to your suggestion, these reasons have been added to improve our description. the revised sentences read as follows: “However, this invasion by Spartina alterniflora was shown to reduce soil respiration, which was contrary to other observations, likely because soil respiration depends on the content and input quality of soil total organic carbon, the decomposition process of litter, or the fluctuation of the groundwater level in wetlands [16-19]” (Line 68)

Point 6: In methods, lines -94- “this habitat was invaded by S. canadensis over the past years.” Pay attention to tenses, and change was to have been.

Response 6: Thank you for the comment. We have revised this sentence, and now the revised sentences read as follows: “however, this habitat has been invaded by S. canadensis over the past years.” (Line 103)

Point 7: In methods, lines -97- “soil was collected from a non-S. canadensis invasion green space at the campus of Jiangsu University.” Pay attention to tenses, and change invasion to invaded.

Response 7: Thank you for the comment. We have revised this sentence, and now the revised sentences read as follows: “soil was collected from a non-S. canadensis invaded green space at the campus of Jiangsu University” (Line 106)

Point 8: In methods, lines -110- “During the experiment period, the soil moisture among three water level treatments were significant variation (P < 0.01; Fig. S1).” Need revise.

Response 8: Thank you for the comment. We have revised this sentence, and now the revised sentences read as follows: “During the experiment period, the soil moisture among three water level treatments were significant difference” (Line 116)

Point 9: In methods, lines -119- While it could be found in previous study, a brief introduction will also be helpful here. Was plant included in the measurement?

Response 9: We appreciate your constructive comments and suggestions. We have added a simple description of the measurement method to make it easier for readers to understand the operation. The plant was not existed in the closed-chamber. The added sentences read as follow: “In brief, the chamber was directly inserted into the soil after removing the weeds, and the carbon dioxide content in the chamber was recorded every 5 seconds for 300 seconds.” (Line 125)

Point 10: Results, -Table 2 - ANCOVA? Is there df for each effect?

Response 10: Thank you for your comments. Yes, we used ANCOVA at here to analyses the univariate relationship between soil respiration and soil moisture under S. canadensis invasion. As your suggestion, we have added the df in the Table 2, which now as follows:

.

S. canadensis invasion

Water level

S. canadensis invasion * Soil moisture

df

F

P

df

F

P

df

F

P

RS

3

0.26

0.62

3

11.92

** 1

9

7.8

**

RA

3

1.88

0.18

3

30.22

**

9

2.55

0.12

RH

3

2.86

0.09

3

19.16

**

9

3.11

0.08

Point 11: Results, -Figure 3 - The P value of the model of each analysis need further check. All of them smaller than 0.05?

Response 11: Thank you for your response. We checked the P value in Figure 3 again, and P value were 0.045, 0.025, and 0.016 in (a), (c), and (e), respectively.

Point 12: Discussion, -240 - “4.1 Effects of S. canadensis invasion on soil total, and heterotrophic, and autotrophic respiration.” And should be removed from the title.

Response 12: Thank you for the comment. We have removed the excrescent word “and” from the subheading, and now the revised sentences read as follows: “4.1. Effects of S. canadensis invasion on soil total, heterotrophic, and autotrophic respiration” (Line 248)

Point 13: Conclusion, lines -346- This sentence could be removed in Conclusion.

Response 13: Thank you for your comments. According to your suggestion, I have deleted this sentence.

Point 14: Conclusion, lines -348- S. canadensis invasion significantly reduced total soil, autotrophic, and heterotrophic respiration. This sentence should be added: "soil".

Response 14: Thank you for the comment. According to your and other reviewers’ suggestion, we have revised this sentence, and now the revised sentences read as follows: “Solidago canadensis L. invasion significantly reduced soil respiration, and the inhibitory effect on autotrophic respiration was stronger than that on heterotrophic respiration.” (Line 353)

Point 15: Conclusion, lines -354- activities but may also affect soil heterotrophic respiration. This sentence should delete “may”.

Response 15: Thank you for the comment. We have removed the word “may” from the sentence, and now the revised sentences read as follows: “, but also affect soil heterotrophic respiration. As soil moisture can be used as a solvent and mobile carrier of soil nutrients,” (Line 359)

Point 16: Conclusion, lines -355- In Conclusion, it is not necessary to expand too much on the potential mechanisms, especially that are not measured in the study. please consider to removed this speculation.

Response 16: Thank you for your comments. According to your suggestion, we have deleted this sentence.

Reviewer 2 Report

Dear Authors

Reviewer report:

Regarding the manuscript entitled “Canada goldenrod invasion regulates the effects of soil moisture on soil respiration” with Manuscript ID ijerph-1986266.

            The study investigates the effects of different degrees of invasion by the worst invasive plant species, Solidago canadensis, on soil respiration under different moisture conditions. The topic of the manuscript is important and the manuscript has interesting results. The paper is written well and the objectives are clear.

The introduction is written well with clear objectives. Materials and methods are written properly with enough information. The results are well written as well as Figures and tables are well presented. The conclusion presents the significance of the results of the present study, however, a recommendation from your data is missed. The supplementary materials are missing. Therefore, I recommend accepting the article after minor revisions for this manuscript.

 I proposed some suggestions that could improve the manuscript

·        Abstract: Please, provide a recommendation from your data regarding the invasion by Canada goldenrod

·        Line 68: “S. canadensis is native to North America and is currently considered one of the most deleterious and pervasive invasive species worldwide” better not start with abbreviation.

·        Line 134: “Plants were harvested on December 15, 2019. The harvested plants were weighed” please, justify at what level? Does the above soil? Or higher? How they are cutted?

·        Line 138: “2.4. Soil characteristics” how is soil collected? Justify.

·        Line 142: “Li et al. (2020) [36].” Adjust.

·        Line 171: “S. canadensis invasion significantly affected soil total, heterotrophic, and autotrophic 171 respiration (P < 0.01, each), and water level significantly affected soil total (P < 0.05) and 172 autotrophic respiration (P < 0.05).” better not start the sentence with an abbreviation.

·        Line 199: Figure 1. I propose using colors instead of patterns for more clear presentation. Also, the resolution of the figure is not good.

·        Figure 3: the resolution is low and the font is hard to follow.

·        Line 251: “S. canadensis was found to have different” start with full name.

·        Line: 362: The supplementary materials are missing. Please, provide them.

·        Line 389: justify.

·        Line 409: “Acacia longifolia” must be italic.

·        Line 438: “Potential Distribution of Goldenrod (Solidago altissima L.) during Climate Change” justify in small letters.

·         

Sincerely Yours,

Author Response

We appreciate all your valuable comments and suggestions to improve the quality of our manuscript. The manuscript has been revised in accordance with you and other reviews' suggestions. Please refer to the Response to Comments file/section below. We hope that the revised manuscript will be acceptable to you.

Point 1: Abstract: Please, provide a recommendation from your data regarding the invasion by Canada goldenrod.

Response 1: Thank you for the constructive comment. We have added the implication from our data in the abstract as follows: “Our results implied that S. canadensis invasion could reduce the soil respiration rate, which further potentially affect the carbon sequestration in the riparian wetlands.” (Line 29)

Point 2: Introduction, lines -68- “S. canadensis is native to North America and is currently considered one of the most deleterious and pervasive invasive species worldwide” better not start with abbreviation.

Response 2: Thank you for your suggestion. We have changed the abbreviation to full name, which now reads as follows: “Solidago canadensis L. is native to North America and is currently considered one of the most deleterious and pervasive invasive species worldwide” better not start with abbreviation.” (Line 74)

Point 3: In methods, lines -134- “Plants were harvested on December 15, 2019. The harvested plants were weighed” please, justify at what level? Does the above soil? Or higher? How they are cutted?

Response 3: Thank you for your comments. In this experiment, all plant tissues were collected, including aboveground parts and underground parts. The cutting position is uniformly divided into stem, leaf and root according to the 0.5cm above the uppermost fibrous root.

Point 4: In methods, lines -138- “2.4. Soil characteristics” how is soil collected? Justify.

Response 4: Thank you for your comments. Soil samples (top soil, 0–10 cm depth) were collected from different points along an X-shaped pattern in in the non-invasive, moderately invasive and completely invasive sample plots under different water level treatment conditions (low water level, medium water level, high water level) using a soil corer (2.0 cm diameter) on December 15, 2019. Soil samples from each pot were mixed thoroughly to obtain one composite soil sample, and all composite soil samples were passed through a sieve (2 mm) to remove visible plant debris and stones and to homogenize before subdividing the samples for analyses. According to your and other reviewers’ suggestions and comments, we have revised the related sentences, which now reads as follows: “Soil samples were collected by mixing topsoil (0–10 cm) obtained from different points along an X-shaped pattern in non-invasive, intermediate invasive, and completely invasive stage treatment pots among all water level treatments on June 15, 2019 (experiment start date) and December 15, 2019 (experiment end date).” (Line 135)

Point 5: In methods, lines -142- Line 142: “Li et al. (2020) [36].” Adjust.

Response 5: Thank you for the comment. According to your comment, we have this sentence, which now reads as follows: “Soil dissolved organic carbon and nitrogen content were quantified using a total organic carbon analyzer with nitrogen module (Shimadzu TOC-L, Kyoto, Japan)” (Line 150)

Point 6: Results, lines -171- “S. canadensis invasion significantly affected soil total, heterotrophic, and autotrophic 171 respiration (P < 0.01, each), and water level significantly affected soil total (P < 0.05) and 172 autotrophic respiration (P < 0.05).” better not start the sentence with an abbreviation.

Response 6: Thank you for your suggestion. we have changed the abbreviation to full name, which now reads as follows: “Solidago canadensis L. invasion significantly affected soil total, heterotrophic, and autotrophic respiration (P < 0.01, each), and water level significantly affected soil total (P < 0.05) and autotrophic respiration (P < 0.05).” better not start the sentence with an abbreviation.” (Line 178)

Point 7: Results, lines -199- Figure 1. I propose using colors instead of patterns for more clear presentation. Also, the resolution of the figure is not good.

Response 7: Thank you for your suggestions on modification. we also apply different colors to distinguish different patterns, so that different categories can be better distinguished whether it is color printing or black and white printing.

Point 8: Results, Figure 3: the resolution is low and the font is hard to follow.

Response 8: Thank you for your reminder. we have improved the picture pixels. I hope you can see it more clearly.

Point 9: Discussion, lines -251- “S. canadensis was found to have different” start with full name.

Response 9: Thank you for your suggestion. We have changed the abbreviation to full name, which now reads as follow: “Solidago canadensis L. was found to have different driving mechanisms for soil total, autotrophic, and heterotrophic respiration.” (Line 258)

Point 10: Supplementary Materials, lines -362- The supplementary materials are missing. Please, provide them.

Response 10: Thank you for the comment. According to the guidelines and requirements of the Journal, the supplementary materials were separated from main text and uploaded. And it is free to readers to download them.

Point 11: References, lines -389- justify. lines -409- “Acacia longifolia” must be italic. lines -438- “Potential Distribution of Goldenrod (Solidago altissima L.) during Climate Change” justify in small letters.

Response 11: Thank you for your suggestion. We have modified the font format and spelling, which now read as follows: “Marchante E, Kjøller A, Struwe S, et al. Short-and long-term impacts of Acacia longifolia invasion on the below-ground processes of a Mediterranean coastal dune ecosystem. Appl. Soil Ecol. 2008, 40(2): 210-217.” (Line 412) and “Park J S, Choi D, Kim Y. Potential distribution of goldenrod (Solidago altissima L.) during climate change in South Korea. Sustainability, 2020, 12(17).” (Line 447)

Reviewer 3 Report

Please find comments, suggestions, and questions on an attachment file. Certain parts are marked with question marks which authors have to re-check before re-submission. Regards,

Author Response

We appreciate your valuable comments and suggestions to improve the quality of our manuscript. The manuscript has been revised in accordance with your suggestions. According to your and two reviewers' comments, revisions have been made in the Introduction, Materials and methods, Discussion and Reference sections in the revised version of the manuscript. We hope that the revised manuscript will be acceptable to you. 

Point 1: Keywords, lines -31- The keyword "Solidago canadensis L." can be deleted.

Response 1: Thank you for your comment. We have deleted this keyword.

Point 2: Introduction, lines -62- This suggests that the alien plant invasive effects on the soil carbon cycle depend on various biotic and abiotic factors. Pay attention to the part of speech, and change "invasive" to "invasion".

Response 2: Thank you for your comment. We have changed the words “the alien plant invasive” to “the alien plant invasion” in the sentence, which now as follows: “This suggests that the alien plant invasion effects on the soil carbon cycle depend on various biotic and abiotic factors.” (Line 68)

Point 3: Introduction, lines -85- The soil total respiration is a kind of soil respiration?

Response 3: We appreciate your valuable suggestion to improve the quality of our manuscript. There are two types of soil respiration which are autotrophic respiration and heterotrophic respiration. Here, the “soil total respiration” means the soil respiration (autotrophic respiration + heterotrophic respiration). We have revised the confusing sentence to make more clear to readers, which now reads as follow: “S. canadensis invasion would inhibit soil respiration, as well as all components of soil respiration (autotrophic and heterotrophic respiration)” (Line 92)

Point 4: In methods, lines -95- There is no result of P.australis stated of result section? And there is no discussion information between P.australis and S.canadensis. Could you please explain, why? And, could you provide that information?

Response 4: We appreciate your constructive comments and suggestions and have considered your suggestions. In this paper, NI is non-intrusive processing, in other words, it is completely a control plot with only P.australis. We have supplemented the contents of relevant P.australis in the results and discussion section, which now read as follows:“Compared with the NI (only P.australis existence) treatment, at low water levels, S. canadensis growth reduced soil total respiration and heterotrophic respiration in the EI, II, DI, and CI stages by 14.3% and 18.0%, 31.7% and 32.7%, 13.1% and 33.6%, 8.5%, and 34.9%, respectively; soil autotrophic respiration was reduced by 4.7% and 32.0% in the EI and II treatments, but was increased by 36.2% and 60.6% in the DI and CI treatments, respectively (P < 0.01, each).” (Line 187) and “Compared with native plants (P.australis), alien invasion plants generally emerge high growth rates, high biomass production, and strong reproductive capacity, which affect the quality and quantity of soil carbon input.” (Line 249)

Point 5: In methods, lines -124- Soil total, heterotrophic and autotrophic respiration were calculated following the equations described in the previous study. Pay attention to punctuation.

Response 5: Thank you for the comment. According to your suggestion, we have add punctuation, which now read as follows: “Soil total, heterotrophic, and autotrophic respiration were calculated following the equations described in the previous study.” (Line 132)

Point 6: Result, lines -127- “Soil samples were collection by mixing topsoil (0–10 cm).” Pay attention tenses. Change collection to collected.

 Response 6: Thank you for your advice. We have changed the words “collection” to “collected” in the sentence, which now as follow: “Soil samples were collected by mixing topsoil (0–10 cm) obtained from different points along an X-shaped pattern in non-invasive.” (Line 135)

Point 7: In methods, lines -129- Please fill in the specific time and pay attention to the unit.

Response 7: Thank you for the comment. According to your suggestion, we have modified the time and unit, which now as follow: “Soil samples were collected by mixing topsoil (0–10 cm) obtained from different points in non-invasive, intermediate invasive, and completely invasive stage treatment pots among all water level treatments on June 15, 2019 (experiment start date) and December 15, 2019 (experiment end date).”  (Line 137) and “Plants were harvested on December 15, 2019. The harvested plants were weighed after 72 hrs oven-drying at 65 °C.” (Line 143)

Point 8: Result, lines -161- “The variations of soil characteristics between start time sampling and end time sampling were used.” Time should be changed to date.

Response 8: Thank you for your advice. We have changed the words “time” to “date” in the sentence, which now as follow: “The variations of soil characteristics between start date sampling and end date sampling were used.” (Line 168)

Point 9: Result, lines -206- “It is suggested to delete the redundant "respiration" to make the sentence more concise.

Response 9: Thank you for your advice. We have revised the superfluous words, which now as follow: “Soil moisture was negatively correlated with soil total, heterotrophic, and autotrophic respiration (P < 0.01, each).” (Line 213), “and plant biomass and ultimately affected soil total, heterotrophic, and autotrophic respiration.” (Line 228), and “The total effects of plant root biomass (-0.418), CUE (-0.269), and alkaline phosphatase activity (-0.497) on soil total, heterotrophic, and autotrophic respiration were the highest.” (Line 230)

Point 10: Discussion, lines -297- root respiration" This should be placed in information section.

Response 10: Thanks for your comments, we have added root breathing to the description, which now as follows: “This complex biochemical process includes autotrophic (root respiration and root microbial respiration) and heterotrophic respiration (microbial and animal respiration).” (Line 54)

Point 11: Discussion, lines -321- Do we have any previous evidence to supply this possibility? Please add this evidence.

Response 11: Thank you for your question. This description is based on our previous study and other references, We have added references supporting this possibility, which now as follow:“It is possible that interspecific competition between native and invasive species intensifies the restriction of substrate availability on soil respiration when water is relatively abundant and reduces the negative effect of water-related factors to a certain extent [32,33]. (Line 331)

Point 12: Discussion, lines -339- “Under S. canadensis invasion, the aerenchyma of the root system expands. Under the influence of soil moisture, it is necessary to balance plant growth, nutrient and water absorption, and hormones, and high soil moisture conditions promote respiration of the plant root system .“ Where are these information coming from? Please cited or did you mean it is a result of reference NO.57?

Response 12: Thank you very much for your comments. We have supplemented the citations of this sentence, which now as follows: “Under the influence of soil moisture, it is necessary to balance plant growth, nutrient and water absorption, and hormones, and high soil moisture conditions promote respiration of the plant root system [32].” (Line 349)